# Clinical Characteristics and Predictors of In-Hospital Mortality among Older Patients with Acute Heart Failure

**DOI:** 10.3390/jcm11020439

**Published:** 2022-01-15

**Authors:** Giuseppe De Matteis, Marcello Covino, Maria Livia Burzo, Davide Antonio Della Polla, Francesco Franceschi, Alexandre Mebazaa, Giovanni Gambassi

**Affiliations:** 1Division of Internal Medicine, Fondazione Policlinico Universitario A. Gemelli IRCCS, Largo Agostino Gemelli 8, 00168 Rome, Italy; giovanni.gambassi@unicatt.it; 2Emergency Department, Fondazione Policlinico Universitario A. Gemelli IRCCS, 00168 Rome, Italy; macovino@gmail.com (M.C.); davide.dellapolla@gmail.com (D.A.D.P.); francesco.franceschi@unicatt.it (F.F.); 3School of Medicine, Università Cattolica del Sacro Cuore, Largo Francesco Vito 1, 00168 Rome, Italy; 4Emergency Department, Ospedale Generale M.G. Vannini, Istituto Figlie di San Camillo, 00177 Rome, Italy; maliburzo@gmail.com; 5IRCSS Ospedale Pediatrico Bambino Gesù, 00165 Rome, Italy; 6Department of Anesthesiology and Critical Care Medicine, AP-HP, Saint Louis Lariboisière University Hospitals, 75010 Paris, France; alexandre.mebazaa@aphp.fr; 7Dipartimento di Medicina e Chirurgia Traslazionale, Università Cattolica del Sacro Cuore, 00168 Rome, Italy

**Keywords:** heart failure, internal medicine, older patients, pulmonary infections, sepsis, mortality

## Abstract

Acute Heart Failure (AHF)-related hospitalizations and mortality are still high in western countries, especially among older patients. This study aimed to describe the clinical characteristics and predictors of in-hospital mortality of older patients hospitalized with AHF. We conducted a retrospective study including all consecutive patients ≥65 years who were admitted for AHF at a single academic medical center between 1 January 2008 and 31 December 2018. The primary outcome was all-cause, in-hospital mortality. We also analyzed deaths due to cardiovascular (CV) and non-CV causes and compared early in-hospital events. The study included 6930 patients, mean age 81 years, 51% females. The overall mortality rate was 13%. Patients ≥85 years had higher mortality and early death rate than younger patients. Infections were the most common condition precipitating AHF in our cohort, and pneumonia was the most frequent of these. About half of all hospital deaths were due to non-CV causes. After adjusting for confounding factors other than NYHA class at admission, infections were associated with an almost two-fold increased risk of mortality, HR 1.74, 95% CI 1.10–2.71 in patients 65–74 years (*p* = 0.014); HR 1.83, 95% CI 1.34–2.49 in patients 75–84 years (*p* = 0.001); HR 1.74, 95% CI 1.24–2.19 in patients ≥85 years (*p* = 0.001). In conclusion, among older patients with AHF, in-hospital mortality rates increased with increasing age, and infections were associated with an increased risk of in-hospital mortality. In contemporary patients with AHF, along with the treatment of the CV conditions, management should be focused on timely diagnosis and appropriate treatment of non-CV factors, especially pulmonary infections.

## 1. Introduction

Heart Failure (HF) incidences are on the rise among older patients, especially in those over 85 years [1,2,3,4,5], and acutely decompensated HF (AHF) has become the leading cause of hospitalization in western countries [6]. AHF is a complex clinical syndrome associated with high mortality, and it is a significant healthcare burden.

During the past two decades, there have been significant advancements in the treatment of patients with chronic, stable HF. In contrast, very little has changed in the management of AHF, and the rates of hospitalization and mortality remain unacceptably high. In managing patients with AHF, the focus has been on the treatment of cardiac failure, resulting in a decline in mortality due to cardiovascular (CV) causes, but this decline is limited to patients younger than 80 years [7].

In the European Society of Cardiology HF Pilot Survey, over half of the patients with chronic HF are ≥75 years, and three-quarters have at least one non-CV comorbidity [8]. In the U.S., patients admitted to hospitals for HF have had an increasing number of non-CV comorbidities in recent years, with the most significant burden among older patients [8,9]. Associated with the increasing prevalence of non-CV comorbidities, there is a progressively higher rate of in-hospital mortality and non-CV causes, especially infections, currently representing the largest number of hospital HF readmissions worldwide [7,9,10]. 

Despite the high prevalence of elderly and medically complex patients hospitalized for AHF [7,11,12], few studies have extensively explored the in-hospital outcomes of patients ≥85 years and possible differences in cause-specific mortality rates by age subgroups are still unknown [5,13,14,15,16]. Furthermore, there is limited evidence about the prognostic impact of non-CV comorbidities and precipitating factors for in-hospital mortality [7,11].

Thus, our study aimed to describe the clinical characteristics of older patients hospitalized with AHF and identify the predictors of in-hospital mortality, distinguishing deaths due to CV and non-CV causes and comparing early rather than late in-hospital events.

## 2. Materials and Methods

We conducted a retrospective study at the Fondazione Policlinico A. Gemelli IRCCS (Rome, Italy), an academic medical center where the emergency department (ED) has an average annual attendance of approximately 80,000 adult patients.

### 2.1. Study Population

Based on electronic health records, we identified all consecutive patients who presented to the ED for ten years between 1 January 2008 and 31 December 2018. Subsequently, data about all patients 65 years of age and older were extracted.

#### 2.1.1. Case Identification and Categorization

To be included in the study sample, patients needed to have received a definitive diagnosis of AHF and be hospitalized in internal medicine wards. The criteria for identifying the cases included an admission diagnosis of AHF adjudicated by the emergency physician and based on a set of standardized parameters including clinical symptoms, physical examination, laboratory data, and radiological findings. In addition, cases needed to have AHF coded as the primary diagnosis in the discharge record. Diagnoses at hospital discharge were based on ICD-10 codes (International Classification of Disease, 10th revision).

Patients presenting to the ED with AHF due to acute coronary syndromes and requiring catheter-based interventions, those with advanced atrioventricular blocks or cardiac tamponade, those who were otherwise admitted to an intensive care unit (ICU), were excluded from the study. Similarly, patients in the terminal/preterminal stage for whom in-hospital death was expected or a palliative care approach was deemed appropriate were also excluded from the study. For patients with multiple recurrent ED admissions for AHF, only data about the first hospitalization were considered.

The final sample was divided into three age groups: patients 65–74 years, patients 75–84 years, and patients ≥85 years.

#### 2.1.2. Data Source

Data were extracted from a centralized hospital data repository searching for a diagnosis of AHF in each ED electronic health record. The individual patient record was then used to collect all demographic and clinical characteristics, all the data regarding ED presentation, and any event occurring during the hospital stay, including outcomes at discharge.

#### 2.1.3. Patients’ Characteristics

At ED admission, for each patient, we collected vital parameters, clinical symptoms, including dyspnea, peripheral edema, chest pain, and NYHA class. 

#### 2.1.4. Established Comorbid Medical Diagnoses

Electronic health records were used to collect information about any established CV and non-CV comorbid medical diagnoses based on the patient’s prior medical history and the listing in the hospital discharge administrative records. Among CV comorbidities, we considered hypertension, chronic atrial fibrillation (AF), coronary artery disease (CAD), prior coronary revascularization, valve disease, cerebrovascular disease, and the placement of a pacemaker (PMK) or a cardiac defibrillator (ICD). Among the non-CV comorbidities, we included diabetes, chronic obstructive pulmonary disease (COPD), chronic kidney disease, chronic liver disease, cancer, and dementia.

#### 2.1.5. Pharmacological Treatment

At ED admission, all long-term CV medications were recorded. Most specifically, we considered the following therapeutic classes: B-Blockers, angiotensin-converting enzyme inhibitors (ACEIs), angiotensin II receptor blockers (ARBs), Mineralocorticoid Receptor Antagonists (MRAs), Loop diuretics, Digoxin, Calcium Channel Blockers, Antiarrhythmic drugs, Oral anticoagulants, and statins. Any possible inconsistent finding of pharmacological treatment that emerged in the ED was resolved after the admission to the internal medicine ward with additional inquiry and by reviewing existing medical records and actual drug boxes.

#### 2.1.6. Acute Medical Conditions

All acute precipitating medical conditions diagnosed during hospital stay were assessed based on hospital clinical diagnoses and discharge administrative records. 

An acute medical condition was defined as the new onset or acute exacerbation of a preexisting clinical condition that could be considered a precipitating or a contributing factor for the presentation to the ED or that emerged early during the hospital stay. Among acute CV medical conditions, we considered ACS, new-onset AF, uncontrolled blood pressure, acute pulmonary embolism, and acute stroke (ischemic or hemorrhagic). Among acute non-CV medical conditions, we considered infections (including urinary tract infections, sepsis/bloodstream infections, and pulmonary infections), worsening of renal function (WRF), anemia, metabolic (including thyroid function and glucose level), and electrolyte derangements.

Information about all these medical conditions was extracted from the electronic medical records and was confirmed based on hospital discharge administrative records. 

### 2.2. Outcome Measures

The primary outcome of the study was the all-cause in-hospital mortality. In addition, we analyzed separately the occurrence of CV- and non-CV-related deaths and early in-hospital events. 

Based on electronic health records and hospital-based death certificates, the causes of death were distinguished between primarily CV- and non-CV-related. Primarily CV-related events were defined as deaths occurring due to terminal HF and cardiogenic shock, acute myocardial infarction, arrhythmias, acute pulmonary embolism, cardiac tamponade, and acute cerebrovascular disease. Non-CV-related events were defined as deaths occurring due to respiratory failure, severe sepsis/septic shock, renal failure, and bleeding with hemorrhagic shock.

Median time (days) of hospital stay was used to distinguish early and late death events. Length of hospital stay (LOS) was calculated as the time from ED admission to hospital discharge or death.

### 2.3. Statistics

Univariate and multivariate analyses were carried out for the overall study sample and separately for each age group. Categorical variables are reported as absolute number (%) and compared by Chi-square test with Yates correction and Fisher’s exact test, as appropriate. Continuous variables are reported as median (interquartile range) and compared by Mann–Whitney U test at univariate analysis. Comparison of multiple groups was assessed by Kruskal–Wallis ANOVA median test. An association to all-cause, in-hospital death was assessed by a univariate Cox regression analysis. Variables that were statistically significant at univariate analysis were entered into a Cox regression model to identify independent predictors for in-hospital death. The results are expressed as Hazard Ratios (HR) and 95% confidence intervals (CI). 

Since a maximum of twelve variables were entered into each logistic regression model, a minimum of 120 death events over 1200 patients would have been required for a correct factor estimation. Thus, our sample is adequate for parameters estimation. All data were analyzed by SPSS v25^®^ (IBM, Armonk, NY, USA).

## 3. Results

### 3.1. Study Sample

Between 1 January 2008 and 31 December 2018, a total of 11,416 patients were admitted to the ED with a diagnosis of AHF. Among these, 3303 patients did not meet inclusion criteria, and 1183 had missing data, yielding a final study sample of 6930 patients (Appendix A). 

#### 3.1.1. Sociodemographic Characteristics

Baseline characteristics of patients by age groups at ED admission are illustrated in Table 1. The mean age was 81 years. Patients 65–74 years of age represented 21% of the sample, and those 75–84 years of age were the largest group (43%). Patients ≥85 years of age represented 36% of the whole population, but their proportion increased steadily across the years considered, from 26% in 2008 to 40% in 2018.

The study sample was almost equally distributed between men (49%) and women (51%), but women accounted for a progressively greater proportion with advancing age (39% among patients aged 65–74 years, 49% among those 75–84 years, and 59% in the oldest group). 

#### 3.1.2. Clinical Variables

Symptoms at ED presentation were almost entirely accounted for by dyspnea (65%) and peripheral leg edema (25%) without differences across age groups. A minority of patients reported chest pain, and this occurred more frequently among younger patients. Nearly 70% of the patients were classified as NYHA class III, while 20% were on NYHA class II.

#### 3.1.3. Comorbid Medical Diagnoses

More than half of the patients had between two and three additional medical diagnoses, and 39% had four or more (Table 1).

Approximately half of the patients had an established diagnosis of hypertension (53%), CAD (48%), and chronic AF (45%). While the prevalence of hypertension was not different in the three age groups, there were opposing trends for CAD and AF. The prevalence of CAD was lower in patients ≥85 years of age (42% vs. 51% in younger patients), whereas the prevalence of AF increases progressively, affecting over 47% of patients ≥85 years of age. Varying degrees of kidney function impairment were present in 43% of the patients with no difference across age groups. Nearly a third of the patients had a diagnosis of diabetes and COPD. The prevalence of diabetes decreased as age increased (37% among patients in the 65–74 years group vs. 24% among patients 85+ years of age). 

#### 3.1.4. Pharmacological Treatment

Information about pharmacological treatment on ED admission was available for 5520 patients (80% of the sample with no differences across age groups). Loop diuretics were the most commonly prescribed class of medications (77%) with no differences across age groups. Among the medications recommended by the clinical guidelines, ACE-I/ARBs were used by 48% and MRAs by 23% of the patients, with only minor differences by age groups. Overall, nearly 60% of the patients were prescribed a beta-blocker but, among those aged 85 years and above, the proportion was much smaller (53%) relative to younger patients (64% in the 65–74 years group). Only a minority of patients used calcium channel blockers, and 13% were on digoxin. Oral anticoagulants use was lower among patients ≥85 years, and a similar pattern was evident for statins.

#### 3.1.5. Acute Medical Conditions

At least one precipitating or acute medical condition could be identified in nearly 68% of patients, and these are reported in Table 2. Infections were present in over 1 out of 4 patients, and they were progressively more prevalent with increasing age. Altogether, infections were identified as the precipitating acute medical condition for 30% of patients ≥85 years of age relative to 25% and 20% for the 75–84 years and 65–74 years groups, respectively. Of all acute infectious diseases, pneumonia and COPD acute exacerbation accounted for 75%. Patients ≥85 years of age had the highest rate of urinary tract infection (7%), while sepsis/bloodstream infection affected about 4% of patients. Uncontrolled blood pressure readings were documented for over 20% of the patients. New-onset AF was diagnosed in 14% of the patients, and other conditions included WRF (5%), anemia (5%), ACS (4%), and electrolyte disturbances (3%). Pulmonary embolism was rarely diagnosed.

#### 3.1.6. In-Hospital Death

Overall, the median LOS for the entire population was ten days (range: 6–16 days). Over the entire period, 908 patients died, with an overall in-hospital mortality rate of 13%. The mortality rate was 8% in patients 65–74 years, 11% in those 75–84, while nearly 1 out of five patients ≥85 years died (19%) during the hospital stay. In the entire cohort, the cause of death was identified as non-CV in nature in 47% of cases. 

The median LOS for deceased patients was eight days (range: 3–17 days). Among the 908 deceased, 465 patients died within the initial eight days of hospital stay (early deaths), and 443 patients died after eight days from admission (late deaths). Patients ≥85 years of age experienced a much higher rate of early deaths (59%) (Table 3). As depicted in Figure 1, 40% of the events occurring early during hospitalization were due to non-CV causes, and these explained the majority of late in-hospital deaths.

The results of the univariate analysis showed that no single variable was associated with an increased risk of in-hospital death in any age group except for diabetes, infections, and WRF (Table 4).

Independent predictors of mortality after adjusting for potential confounders are reported in Table 5. NYHA class was the only functional parameter that resulted in an independent predictor of in-hospital mortality with graded severity. The finding was consistent across the different age groups, although the association appeared stronger in patients ≥85 years, HR 2.11; 95% CI 1.51–2.93 (*p* < 0.001) for NYHA class III and HR 3.09; 95% CI 2.12–4.50 (*p* < 0.001) for NYHA class IV. Among CV comorbidities, coronary artery disease emerged as a predictor of death in patients <85 years and cerebrovascular disease in patients >85 years. Among non-CV comorbidities, dementia was associated with a higher risk of in-hospital death in patients ≥85 years of age. Considering the acute medical conditions, infections were associated with an almost two-fold increased risk of death in all age groups, HR 1.74 in patients 65–74 years, HR 1.83 in patients 75–84 years, and HR 1.74 in patients ≥85 years. A WRF was similarly associated with a higher risk of in-hospital death in all age groups. In contrast, a new-onset AF and uncontrolled blood pressure and anemia were associated with a reduced risk of mortality in all age groups.

## 4. Discussion

Our study has documented that patients ≥85 years of age are the largest group among AHF patients hospitalized in internal medicine wards. These are complex patients, with a third of them having four or more comorbidities above and beyond HF, with a significantly increased burden of non-CV comorbidities that contribute to an increased risk of hospitalization. In-hospital mortality was 2.5 times higher in patients ≥85 years of age than those 65–74 years, especially during the initial days following hospital admission. The most common precipitating factor for acute decompensation was pulmonary infections, and overall, infections were associated with an almost two-fold increased risk of mortality in all three age groups.

The overall burden of HF is continuously increasing, with individuals ≥75 years old being reported to account for at least half of the patients [17,18]. However, in a “real-world” setting, our study highlights that, in recent years, the majority of patients hospitalized for AHF are indeed those ≥85 years with a mean age of about 90 years. The trend is rising, and in the coming years, there will be a constant increase in AHF hospitalizations of progressively older patients [18]. These patients are rarely studied and are not enrolled in randomized clinical trials. As a consequence, HF in patients ≥85 years remains poorly understood, and there is evidence that the management of these patients does not follow recommended guidelines. Indeed, in the current study, the number of patients ≥85 years and their mean age is higher than that analyzed in previous reports, either in Italy or Europe [5,13,16]. Furthermore, at odds with previous studies based on registries and surveys [5,19], we considered exclusively patients admitted to internal medicine wards as it has been documented that those admitted to cardiology units are a selected group of younger patients with significant differences in clinical characteristics and outcomes, in Italy and elsewhere [20,21]. 

Consistent with the findings of prior studies, AHF patients ≥85 years are most commonly females and have a higher prevalence of hypertension compared with patients 65–74 years [10]. Instead, CAD/prior coronary revascularization and diabetes, along with cancer, show an inverse age relation. The latter is a universal finding due to the increased lethality of these conditions [22]. It is noteworthy that the prevalence of individual CV and non-CV comorbidities are very similar to that previously reported among hospitalized AHF patients in Italy and Europe [5,19]. Interestingly, in our study, we did not observe the decline in prevalence in the very elderly group following a peak among patients 75–84 years, seen previously [16]. This could be because prior studies included patients screened for enrollment in clinical trials or because such decline is shifting to even later in age. This latter hypothesis is substantiated by studies on AHF patients in Europe and the U.S. Veterans Affairs medical system, where the authors have included patients between 80 and over 90 years old [23,24]. This finding raises the possibility that, due to the convergence of several factors, such as improved treatments and a slower aging process, these patients may avoid earlier mortality related to their multiple comorbidities and show progressive longevity. 

Several conditions are recognized to precipitate AHF, and their ascertainment is recommended by current AHF guidelines [25]. In our sample, at least one precipitating factor was identified in over two-thirds of the patients. Interestingly, only 27% of patients ≥85 years had none identified compared with 37% of those 65–74 years of age. These figures are somewhat higher but comparable with findings from other authors. Among 15828 AHF patients (mean age 71 years) enrolled by the GREAT Network, a precipitating factor could be identified in 55%, with the great majority having one single precipitant [26]. Among those with just one precipitating factor, the prevalence of uncontrolled hypertension (11%) and new-onset AF (16%) were similar to our study. In our study, infections, without differentiation between prehospital and in-hospital ones, were identified as a precipitating factor in 26% of the patients, and these findings are concordant with previously reported estimates comprised between 14 and 20% [26,27]. Specifically, among these, pulmonary infections were the most frequent precipitating factor, as noted in a previous review of AHF management in older patients [28]. Acute coronary syndromes, a well-known strong predictor of worse outcomes, were rarely identified as a precipitating factor in our study. This finding is likely due to the exclusion of patients presenting to the ED with AHF due to ACS and requiring admission to specialized cardiology or an intensive care unit. However, it is noteworthy that in the Biomarcoeur cohort of AHF patients admitted to three EDs in France, Tunisia, and Turkey, ACS was deemed a precipitating factor in just 6% of patients [27]. It is noteworthy that precipitating factors of AHF may substantially influence outcomes far beyond the duration of hospitalization, and this suggests opportunities for improved management and prognostication. Such consideration seems of utmost importance concerning pulmonary infections. There is a high 90-day risk of death when an infection precipitates AHF, and the risk is maximal at three weeks from hospital admission [26].

The overall in-hospital mortality rate in our study was 13%. In a similar Italian AHF sample, the all-cause in-hospital death rate was 6.4% but, despite a similar prevalence of comorbidities, the patients were younger (mean age 72 years) and were admitted exclusively in cardiology units [5]. Likewise, among 9999 AHF patients aged 80 years treated in Spanish emergency departments, only 10% of patients died within 30 days of admission to the hospital [11]. Similar in-hospital and 28-day mortality was reported for patients (77 years) in the ARIC Study in the U.S. [29]. These differences are probably due to the heterogeneity of the patients included in the studies. However, in our study, the in-hospital mortality rate was 8–11% for patients 65–74 and 75–84 years, respectively, but it was 2.5-fold higher among patients ≥85 years. A similarly increased mortality rate for HF octogenarians was reported in the IN-HF Outcome study [5] and the Euro Heart Failure Survey I [19]. Compared with patients 81 years of age, mortality rates were highest for those aged 90 years also among US veterans, both during hospitalization and within 30 days from admission [24].

Interestingly, in 47% of cases, the cause of death was adjudicated as non-CV nature, with no statistically significant differences across age groups. This finding appears to be broadly consistent in different populations, heterogeneously classified, and with diverse follow-up times [10,30,31]. Also, a stepwise increase in in-hospital mortality has been documented with a progressively higher number of non-CV comorbidities [9,29]. In addition, a longitudinal analysis of 86,000 patients with incident HF has shown that between 2002 and 2013, all-cause mortality declined among younger patients but not among those ≥80 years of age. Most specifically, in these patients, the reduced CV-related mortality has been entirely offset by a steeply increased number of deaths due to non-CV conditions [7]. Altogether, these findings confirm that non-CV deaths prevention is a crucial therapeutic goal in patients hospitalized for AHF and a challenge to current management strategies. 

Over one-half of all deaths occurred in the immediate few days following hospital admission, and ≥85 years patients had a significantly higher early death rate (59%) compared with younger groups (41–44%). In 4 out of 10 cases, the early death of patients ≥85 years of age was caused by non-CV conditions. We did not collect information about in-hospital treatment, and we could not make judgments regarding possible age-related under-treatment or less aggressive management of precipitating acute medical conditions. Yet, the use of HF and CV-related chronic medications was not differential across groups, and usually, the most appropriate treatments are deployed regardless of chronological age, at least in the initial days of hospital stay. Thus, our findings seem to suggest that a more aggressive and timely treatment, aimed at both CV and non-CV comorbidities and precipitating factors, is warranted to reduce early deaths in these AHF patients. 

After adjusting for potential confounders, a few variables resulted independently associated with in-hospital mortality. As expected, a worse functional NYHA class was consistently associated with higher mortality in all patients, but the strength of the association was greater with increasing age. While coronary artery disease retains an association with in-hospital mortality among patients 65–74 years, cerebrovascular disease and dementia were negative prognostic factors only in the oldest group of patients. All of these findings are consistent with several prior studies [16,32,33]. In contrast, ACS was not a predictor of in-hospital mortality in any age groups considered. This relates to the notion that patients with more-severe ACS and poorer prognosis, not manageable in an internal medicine ward but needing transfer to cardiology or an intensive care unit, were excluded from the study. Furthermore, as stated by the AHF Committee of the HF Association of the European Society of Cardiology, the diagnosis of ACS in the setting of AHF is particularly challenging as the pillars of an ACS diagnosis may be confounded by AHF, and most other AHF studies have excluded many patients with ACS and vice versa [34]. 

In addition, our data show that a new-onset AF, uncontrolled blood pressure, and anemia were associated with a more favorable outcome in all age groups. Indeed, other recent surveys reported a similar finding, which is explained by the possible, timely correction of these precipitating factors [11,26]. However, we cannot conclusively reject that the findings of uncontrolled hypertension may have been influenced by a possible higher prevalence of patients having HF with preserved ejection fraction (EF), which may have justified their survival advantage. Consistently with an updated meta-analysis, WRF was associated with an increased risk of death [35,36,37].

In our study, infections were the main non-CV predictor of an increased in-hospital, all-cause mortality in all age groups. Infections, mainly pulmonary infections, increased the risk of mortality almost two-fold. This finding is consistent with a recent survey, which identified infections not only as one of the most prevalent precipitating factors of death in AHF patients but also as one of the main predictors of non-CV mortality and hospitalizations for HF [7,38]. However, only a few other studies have explored the prognostic impact of infections in patients with HF [10,39], and in some instances, AHF patients with infection were not found at higher risk [11]. In a study of 1802 HF with reduced EF, followed over a mean follow-up period of 4 years, 23% of deaths were due to sepsis, and pulmonary infections accounted for 70% of these events [31]. However, even if infections were the main predictor of increased in-hospital mortality in our analysis, we did not differentiate between prehospital and in-hospital infections. We did, however, consider all infections that acted as precipitating or contributing factors for the presentation to the ED or that emerged early during the hospital stay. A recent study evaluating patients enrolled in the Kyoto congestive HF registry also showed that infections were associated with higher in-hospital and post discharge mortality in patients with AHF. However, the authors considered the association of clinical outcomes with only newly diagnosed infections after hospitalization, excluding patients that presented signs and symptoms of infection on admission [40].

Infection and AHF hospitalizations are associated with significantly worse survival rates than other CV and non-CV hospitalizations, and infection-related mortality is comparable to that of terminal HF. The infection-related death in patients with AHF appears to be higher than in the general population, lending support to the hypothesis that AHF is a possible risk factor for infection-related death [10]. This seems further supported by the notion that infections, especially pulmonary, are independently associated with higher mortality of AHF patients at 90 days and long-term [10,31,39]. In AHF patients, infections often present without classical clinical signs and can be more challenging to diagnose, emphasizing the need for a high suspicion at the time of admission.

Altogether, our results seem to confirm and extend our understanding of the declining mortality observed among contemporary AHF patients. While the improvement in the management of CV diseases has led to a reduction in CV-related deaths, this reduction in mortality is decidedly less steep in octogenarian patients. In fact, the rates of overall mortality for AHF in these patients have not changed due to the increasingly negative impact of non-CV factors, especially infections. 

### Limitations

Although conducted at one of the largest medical centers in Italy, our study may not be representative of all AHF patients. However, in the ARNO Observatory, the profile of AHF patients discharged primarily from internal medicine wards appear to be very similar to the patients in our study [13]. Given the retrospective nature of our study and the fact that all data were extracted from electronic medical records, some relevant clinical variables were not available, and others could have been missed. Most specifically, natriuretic peptides level was not assessed systematically, and no data was available about echocardiographic evaluations and the EF phenotype of these HF patients. The ascertainment of precipitating factors could have been incomplete, and patients without an identified precipitating factor might indeed have unrecorded precipitants. As in prior studies, we could not estimate the relevance of nonadherence and noncompliance to chronic HF medications [4]. However, we focused on a limited list of relevant non-CV precipitating factors as in previous studies [26,27], and selective under-reporting is unlikely. In particular, infections were considered precipitating factors for AHF without differentiating between preexisting and nosocomial. Moreover, we do not have data on specific pathogens in most of our patients, reflecting the challenge of establishing this in many patients with respiratory tract infections as already reported [31]. No information was available about the pharmacological approach, including the frequency of inotrope usage, during the hospitalization. Moreover, we do not have data about the frequency of device use among hypotensive patients, such as intra-aortic balloon pump (IABP) or left ventricular assist device (LVAD), nor the frequency of mechanical ventilator use. Furthermore, we lack detailed information about the treatment of the patients while in the ED. Indeed, in many patients admitted with AHF, the treatment begins in the emergency room, and its timing could impact the prognosis. In this respect, it is noteworthy that patients spent a similar amount of time in the ED. They were hospitalized in internal medicine wards with permanent medical staff under the same coordinating chief, constantly reevaluating implementing clinical guidelines. The specific causes of death were derived from death certificates that may not have always been accurate. Finally, after the discharge of the patients, there was no systematic follow-up, and we could not obtain information about living status nor about hospital readmissions.

## 5. Conclusions

AHF patients are increasingly very old, more frequently females, and have a significant non-CV comorbidities burden. In-hospital mortality is much higher in the oldest patients, and deaths tend to occur early during hospitalization. Overall, non-CV causes account for nearly half of deaths, and infections, especially pulmonary, are the most common predictor of in-hospital excess mortality. Since the hospital admission of these patients will continue to increase, it is imperative that we optimize the management of non-CV conditions and identify strategies to improve outcomes and reduce early in-hospital mortality. Ongoing efforts to improve survival will have to carefully consider prevention, timely diagnosis, and appropriate treatment of non-CV precipitating factors, especially acute infections. Since these contemporary AHF patients will continue to be admitted to internal medicine wards, the high in-hospital mortality associated with infections highlights the need for multidisciplinary team-based care to manage these complex patients.

## Figures and Tables

**Figure 1 jcm-11-00439-f001:**
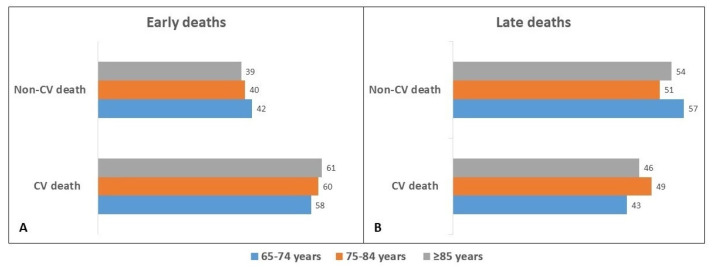
Comparison of Early (**A**) vs. Late (**B**) mortality rates according to causes of death among the different age groups.

**Table 1 jcm-11-00439-t001:** Demographic and clinical characteristics of patients by age groups.

	65–74 Years (*n* = 1464)	75–84 Years (*n* = 2975)	≥85 Years (*n* = 2491)	All (*n* = 6930)	*p* Value
Age (years)	70.2 ± 2.8	79.8 ± 2.8	89.0 ± 3.4	81.1 ± 7.5	<0.001
Female	38.6	49.3	59.1	50.6	<0.001
Clinical symptoms/signs					
Dyspnea	66.4	64.9	64.7	65.1	0.521
Peripheral edema	25.8	25.5	23.8	24.9	0.277
Chest pain	15.8	12.8	9.4	12.2	<0.001
NYHA class					
II	20.2	20.9	18.4	19.9	
III	70.4	68.6	69.5	69.3	0.023
IV	9.4	10.5	12.1	10.8	
Vital parameters					
Systolic blood pressure (mmHg)	138 ± 30	137 ± 29	137 ± 29	137 ± 29	0.538
Diastolic blood pressure (mmHg)	77 ± 17	76 ± 18	75 ± 17	76 ± 17	0.006
Heart Rate (beats/min)	88 ± 23	86 ± 23	86 ± 22	86 ± 23	<0.001
O2 saturation	93 ± 5	93 ± 6	92 ± 6	93 ± 6	<0.001
Number of comorbidities					
0–1	24.2	2.3	2.2	6.9	
2–3	36.0	61.6	55.8	54.1	<0.001
4+	39.8	36.1	42.0	39.0	
CV comorbidities					
Coronary artery disease	51.9	51.1	42.0	48.0	<0.001
Prior coronary revascularization	20.7	17.6	10.6	15.7	<0.001
Hypertension	50.1	54.8	53.7	53.4	0.016
Valve disease	25.6	21.6	18.9	21.5	<0.001
Chronic atrial fibrillation	40.6	45.2	47.3	44.9	<0.001
Pacemaker/ICD	20.2	21.2	19.2	20.3	0.190
Cerebrovascular disease	6.1	8.9	8.1	8.1	0.004
Non-CV comorbidities					
Diabetes	37.1	32.9	23.5	30.4	<0.001
Chronic obstructive pulmonary disease	25.3	32.2	31.4	30.4	<0.001
Chronic kidney disease	42.8	44.3	41.5	43.0	0.208
Chronic Liver disease	3.8	2.1	0.8	2.0	<0.001
Cancer	11.7	9.1	7.7	9.2	0.021
Dementia	0.6	4.0	8.4	4.8	<0.001
Medications					
Loop Diuretics	75.3	77.7	76.5	76.8	0.097
ACE-I/ARBs	47.9	49.3	45.8	47.7	0.145
β-Blockers	64.4	61.5	53.1	59.0	<0.001
Mineralocorticoid antagonists	28.2	22.8	21.5	23.4	0.001
Digoxin	12.5	12.7	14.1	13.2	0.214
Calcium Channel Blockers	14.6	17.2	16.0	16.2	0.286
Antiarrhythmic drugs	14.0	13.4	10.8	12.6	0.018
Oral anticoagulants	41.4	40.1	31.9	37.4	<0.001
Statins	30.6	29.3	17.2	25.2	<0.001

Values are given as % for categorical variables or as means ± standard deviations for continuous variables. Abbreviations: NYHA = New York Heart Association; CV = cardiovascular; ICD = implantable cardioverter-defibrillator; ACE-I = angiotensin-converting enzyme inhibitors; ARBs = angiotensin receptor blockers.

**Table 2 jcm-11-00439-t002:** Precipitating factors and acute medical conditions by age groups.

	65–74 Years (*n* = 1464)	75–84 Years (*n* = 2975)	≥85 Years (*n* = 2491)	All (*n* = 6930)	*p* Value
Precipitating factors					
At least one	63.5	65.8	72.7	67.8	<0.001
Not identified	36.5	34.2	27.3	32.2	
Acute medical conditions					
Uncontrolled blood pressure	20.4	20.0	21.4	20.6	0.438
Acute coronary syndrome	3.2	3.2	4.0	3.5	0.227
Acute pulmonary embolism	0.8	0.8	1.2	0.9	0.245
New-onset atrial fibrillation	14.5	13.4	13.6	13.7	0.115
Stroke	1.6	2.8	3.7	2.9	0.001
Fever/febrile episodes	5.5	5	5.3	5.3	0.766
Infections	20.2	25.4	29.9	25.9	<0.001
Pneumonia	9.7	11.1	17.3	13.1	<0.001
COPD acute exacerbation	4.6	7.0	6.3	6.2	0.011
Urinary tract infection	2.7	3.9	7.0	4.8	<0.001
Sepsis/bloodstream infection	3.3	4.2	3.5	3.8	0.264
Worsening renal function	4.1	4.5	6.1	5.0	0.005
Metabolic/Electrolyte derangement	2.3	2.4	4.1	3.0	<0.001
Anemia	5.4	5.3	5.1	5.3	0.905

Values are given as % for categorical variables. Abbreviations: COPD = Chronic obstructive pulmonary disease.

**Table 3 jcm-11-00439-t003:** In-hospital length of stay and mortality.

	65–74 Years (*n* = 1464)	75–84 Years (*n* = 2975)	≥85 Years (*n* = 2491)	All (*n* = 6930)	*p*
Length of stay (days)	11 (7–18)	10 (7–17)	9 (6–16)	10 (6–16)	<0.001
All-cause In-hospital mortality	8	11	19	13	<0.001
Causes of death					
CV-related	49.3	53.9	54.7	53.4	0.541
Non-CV related	50.7	46.1	45.3	46.6	
CV/Non-CV ratio	0.97	1.16	1.20	1.14	
Time of death					
Early deaths	41.4	44.2	59.1	51.2	<0.001
Non-CV related	41.6	39.7	39.1	39.6	
Late deaths	58.6	55.8	40.9	48.8	
Non-CV related	57.3	51.3	53.8	53.3	

Values are given as median and interquartile ranges (IQR) for continuous variables and as % for categorical variables. Abbreviations: CV = cardiovascular.

**Table 4 jcm-11-00439-t004:** Predictors of in-hospital mortality by age groups (Univariate analysis).

	65–74 Years		75–84 Years		≥85 Years		All	
	Dead/Alive (116/1348)	*p*	Dead/Alive (342/2633)	*p*	Dead/Alive (450/2041)	*p*	Dead/Alive (908/6022)	*p*
Age (deceased)	71 (68–73)	0.369	80 (78–83)	0.012	88 (87–92)	0.079	84 (79–88)	<0.001
Female	37.9	0.879	45.9	0.181	59.1	0.992	51.4	0.581
NYHA class								
II	14.7		14.6		8.9		11.8	
III	69.0	0.015	67.3	<0.001	71.8	<0.001	69.7	<0.001
IV	16.4		18.1		19.3		18.5	
Number of comorbidities								
0–1 2–3 4+	19.8 44.0 36.2	0.162	11.7 51.8 36.5	<0.001	11.1 50.4 38.4	<0.001	12.4 50.1 37.4	<0.001
CV comorbidities								
Coronary artery disease	66.4	0.001	57.9	0.007	44.2	0.281	52.2	0.006
Hypertension	58.6	0.063	57.3	0.339	57.8	0.059	57.7	0.007
Valve disease	28.4	0.547	27.2	0.036	22.7	0.142	25.1	0.044
Chronic atrial fibrillation	47.4	0.118	49.1	0.119	52.7	0.011	50.7	<0.001
Pacemaker/ICD	24.1	0.273	19.6	0.445	16.0	0.058	18.4	0.133
Cerebrovascular disease	5.2	0.670	12.3	0.020	11.6	0.007	11.0	0.001
Non-CV comorbidities								
Diabetes	48.3	0.009	38.0	0.032	35.8	<0.001	38.2	<0.001
COPD	23.3	0.594	29.5	0.261	28.9	0.213	28.4	0.153
Chronic kidney disease	47.4	0.369	46.2	0.686	42.0	0.679	44.3	0.850
Chronic Liver Disease	1.7	0.230	3.2	0.134	2.7	<0.001	2.8	0.078
Cancer	22.4	0.001	14.0	0.020	9.8	0.816	13.0	0.022
Dementia	1.7	0.111	6.4	0.015	12.7	<0.001	8.9	<0.001
Acute medical conditions								
Uncontrolled blood pressure	10.3	0.005	9.6	<0.001	13.8	<0.001	11.8	<0.001
Acute coronary syndromes	4.3	0.484	3.8	0.523	3.6	0.584	3.7	0.676
Acute pulmonary embolism	0.9	0.598	0.9	0.742	1.6	0.463	1.2	0.303
New onset atrial fibrillation	6.9	<0.001	11.4	<0.001	16.0	<0.001	13.1	<0.001
Syncope	5.2	0.749	3.8	0.742	3.3	0.217	3.7	0.353
Stroke	1.7	0.703	2.0	0.485	4.0	0.703	3.0	0.821
Fever/febrile episodes	7.8	0.274	5.8	0.469	5.6	0.822	5.9	0.314
Infections	31.9	0.001	34.5	<0.001	36.4	0.001	35.1	<0.001
Worsening of renal function	19.0	<0.001	13.2	<0.001	12.0	<0.001	13.3	<0.001
Metabolic/Electrolyte disturbances	1.7	1.000	0.9	0.057	5.1	0.209	3.1	0.811
Anemia	3.4	0.517	2.6	0.019	3.1	0.034	3.0	0.001

Values are given as median and interquartile ranges (IQR) for continuous variable and as % for categorical variables. Abbreviations: NYHA = New York Heart Association; CV = cardiovascular; ICD = implantable cardioverter-defibrillator; COPD = Chronic obstructive pulmonary disease.

**Table 5 jcm-11-00439-t005:** Predictors of in-hospital mortality in different age groups (Multivariate analysis).

	65–74 Years		75–84 Years		≥85 Years		All	
Variable	HR (95% CI)	*p*	HR (95% CI)	*p*	HR (95% CI)	*p*	HR (95% CI)	*p*
Age							1.08 (1.07–1.09)	<0.001
NYHA class II	Ref.		Ref.		Ref.		Ref.	
NYHA class III	1.64 (1.20–2.24)	0.002	1.64 (1.20–2.24)	0.002	2.11 (1.51–2.93)	<0.001	1.80 (1.46–2.21)	<0.001
NYHA class IV	2.44 (1.67–3.56)	<0.001	2.43 (1.67–3.56)	<0.001	3.09 (2.12–4.50)	<0.001	2.73 (2.14–3.49)	<0.001
Coronary artery disease	1.42 (1.13–1.78)	0.002	1.42 (1.14–1.77)	0.002			1.38 (1.19–1.58)	<0.001
Hypertension							1.38 (1.20–1.59)	<0.001
Valve disease			1.18 (0.93–1.51)	0.173			1.28 (1.09–1.49)	0.002
Cerebrovascular disease			1.30 (0.94–1.80)	0.120	1.53 (1.14–2.04]	0.004	1.42 (1.15–1.75)	0.001
Diabetes	1.13 (0.91–1.42)	0.274	1.13 (0.91–1.42)	0.274	1.44 (0.82–1.94)	0.453	1.55 (1.34–1.79)	<0.001
Cancer	1.48 (1.09–2.03)	0.013	1.48 (1.09–2.03)	0.013	…	…	…	…
Dementia	...	...	1.74 (1.12–2.71)	0.014	1.74 (1.43–2.12)	<0.001	1.59 (1.25–2.02)	<0.001
Infections	1.74 (1.10–2.71)	0.014	1.83 (1.34–2.49)	0.001	1.74 (1.24–2.19)	0.001	1.97 (1.61–2.41)	0.001
New onset atrial fibrillation	0.61 (0.44–0.86)	0.005	0.61 (0.43–0.86)	0.005	0.68 (0.53–0.88)	0.003	0.76 (0.59–0.86)	0.002
Uncontrolled blood pressure	0.49 (0.26–0.88)	0.018	0.44 (0.31–0.64)	<0.001	0.58 (0.43–0.74)	<0.001	0.50 (0.41–0.62)	<0.001
Worsening renal function	2.11 (1.53–2.92)	<0.001	2.11 (1.53–2.92)	<0.001	1.92 (1.44–2.56)	<0.001	2.21 (1.82–2.69)	<0.001
Anemia			0.46 (0.24–0.90)	0.023	0.59 (0.34–1.02)	0.051	0.63 (0.43–0.93)	0.019

The results are expressed as Hazard Ratios (HR) and 95% confidence intervals (CI). Abbreviations: NYHA, New York Heart Association.

## Data Availability

The data presented in this study are available on request from the corresponding author.

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
