# Peer review of "Clinical Characteristics and Predictors of In-Hospital Mortality among Older Patients with Acute Heart Failure"

_jcm, 2022, doi:10.3390/jcm11020439_

Round 1

Reviewer 1 Report

I have read the manuscript entitled 'CLINICAL CHARACTERISTICS AND PREDICTORS OF IN- 2 HOSPITAL MORTALITY AMONG OLDER PATIENTS WITH 3 ACUTE HEART FAILURE'
The study appears to be well written and presented. I have several minor concerns regarding the article. 
First of all, the manuscript should be checked for typo errors since there are some.
The authors should explain the frequency of inotrope medical usage among their study population. Do they need any assist devices among hypotensive patients such as IABP or LVAD. The frequency of mechanical ventilator use should be added. If they do not have the aforementioned data, they should be added to the limitation section.
Worsening renal functions are very important in these patients with acute heart failure which should be further discussed. The long -term effect of worsening renal functions have also been reported. Please consider citing  more recent articles such as 'Effect of acute kidney injury on long-term mortality in patients with ST-segment elevation myocardial infarction complicated by cardiogenic shock who underwent primary per'cutaneous coronary intervention in a high-volume tertiary center' and 'Clinical characteristics and outcomes of acute coronary syndrome patients with intra-aortic balloon pump inserted in intensive cardiac care unit of a tertiary clinic'

Reviewer 2 Report

In this study, the authors investigated an over 65 year old AHF population  in a single centre database. They found that ~30% AHF patients experienced hospitalization due to infection, and that the risk of death was increased two-fold in infected AHF patients.  The strategy looks sound, and the results are reasonable. Moreover, this is wonderful to see they have a variety of variables. I would like to raise a couple of issues listed below.

Major:

  1. Please state definitions of comorbid medical diagnoses and acute medical function
  2. As hospitalized AHF patients are susceptible to nosocomial infections. It may affect up to 10% of AHF patients [1]. It is important to explain what determined infection to be called the AHF driver and distinguish it from nosocomial infections. How was the differentiation between prehospital and inhospital infection made?
  3. Since anemia is common and affects about 10-50% of HF population it is considered more of a chronic rather than acute state [2]. The authors defined anemia as an acute condition and reported its rate as low as ca 3%. Was based seldom on Hgb level? Or were there other accompanying factors supporting anemia as an acute condition such as bleeding. Please explain.

Minor

1) Please add to tables’ heading that they refer to inhospital mortality  (Table 4,5)

2) It would be of great value to have information regarding HF fenotypes ( preserved, mr and redcued HF). However, this was already noted and reported in limitations of the study.

3) It would be of value if authors could  assess relation between comorbidities and risk of infection. Is there a greater risk of infection in patiens with greater comorbidity burden?

1 Seko Y, Kato T, Morimoto T, Yaku H, Inuzuka Y, Tamaki Y, Ozasa N, Shiba M, Yamamoto E, Yoshikawa Y, Yamashita Y, Kitai T, Taniguchi R, Iguchi M, Nagao K, Jinnai T, Komasa A, Nishikawa R, Kawase Y, Morinaga T, Toyofuku M, Furukawa Y, Ando K, Kadota K, Sato Y, Kuwahara K, Kimura T; KCHF Study Investigators. Newly Diagnosed Infection After Admission for Acute Heart Failure: From the KCHF Registry. J Am Heart Assoc. 2021 Nov 16;10(22):e023256. doi: 10.1161/JAHA.121.023256. Epub 2021 Nov 3. PMID: 34730004.

  1. 2 Anand IS, Gupta P. Anemia and iron deficiency in heart failure current concepts and emerging therapies. Circulation 2018;138:80–98.

Reviewer 3 Report

Thank you for the up to date work.

It would be interesting to know the authors's viewpoint concerning the perspectives of further studying of different types of HF and preventive measures among older patients connected with these results.

Round 2

Reviewer 2 Report

Dear Authors.

Thank you for your detailed replies and corrections to all reviewers.  

The  only concern that remains regards clarity in terms of relation between infection and AHF. I find abstract and introduction clear, however, later on in RESULTS and DISCUSSION it starts to be confusing as infection is mainly described as a AHF precipitating factors and the fact this could be nosocomial infections is missed.

As authors replied infections reported in the study include both community and hospital-acquired infections. So my request is to make it clear to the reader that both types are included as it affects data interpretation.

Please note that the cited article by Seko  et al states clearly that newly diagnosed infection were taken into account while patients presenting signs and symptoms of infection on admission were excluded.

I would suggest to makes correction in the following parts to explain that infections reported could be both factor precipitating AHF in the prehospital phase or factor complicating AHF that already occurred

  • In the section 3.1.5 my suggestion is to rewrite line 216 to “At least one precipitating or acute medical condition”
  • Discussion lines: 288, 331
  • Discussion 417-424 In this part, I would suggest to underline the fact preexsing and new infections are taken into account in the authors study.

I would be of value to add in limitations that some acute conditions for instance infection or anemia could be preexisting thus potentially precipitating AHF or could be related to hospitalization, for instance nosocomial infections, bleedings.

Thank you 
